# A Survey of Synthetic Routes and Antitumor Activities for Benzo[*g*]quinoxaline-5,10-diones

**DOI:** 10.3390/molecules25245922

**Published:** 2020-12-14

**Authors:** Alain G. Giuglio-Tonolo, Christophe Curti, Thierry Terme, Patrice Vanelle

**Affiliations:** Equipe Pharmaco-Chimie Radicalaire, Faculté de Pharmacie, Institut de Chimie Radicalaire, Aix Marseille Université, CNRS, UMR 7273, 27 Boulevard Jean Moulin, CEDEX 05, 13385 Marseille, France; christophe.curti@univ-amu.fr (C.C.); thierry.terme@univ-amu.fr (T.T.)

**Keywords:** benzo[*g*]quinoxaline-5,10-dione, 1,4-diazaanthraquinone, anthracene-9,10-dione

## Abstract

Anthracycline antibiotics play an important role in cancer chemotherapy. The need to improve their therapeutic index has stimulated an ongoing search for anthracycline analogs with enhanced properties. This review aims to summarize the common synthetic approaches to benzo[*g*]quinoxaline-5,10-diones and their uses in heterocyclic chemistry. Because of the valuable biological activities of the 1,4-diazaanthraquinone compounds, a summary of the most promising heterocyclic quinones is provided together with their antitumor properties.

## 1. Introduction

The substructure of 1,4-anthraquinones (anthracene-9,10-diones) is an important class of bioactive non-heterocyclic quinonoid compounds (Figure 1). They include natural anthracyclines such as Daunomycin, mainly active against acute lymphoblastic leukemia and acute myeloid leukemia, and Doxorubicin approved for the treatment of a wide variety of liquid and solid tumors [1]. The reference synthetic 1,4-anthraquinone is Mitoxantrone, with anticancer applications but also active on multiple sclerosis through its immunosuppressant properties [2]. Both drugs have adverse side effects typical of non-selective cytotoxic drugs, with inhibitor effects on rapidly dividing tissues (hair, bone marrow and mucous membranes). Moreover, 1,4-anthraquinones produce a dose-limiting specific toxicity to the heart. This has led to the development of many analogs with reduced toxicity and improved spectrum of activity [3]. Recently pixantrone, an aza-anthracenedione, was developed to reduce cardiotoxicity typically associated with anthracyclines but without compromising antineoplastic efficacy [4].

Numerous quinones are known for their anticancer activities, such as the alkylating agent Mitomycine C [5], or Streptonigrin obtained from *Streptomyces flocculus*, whose application is limited by its toxicity [6].

1,4-Diazaanthraquiones exhibit promising in vitro and in vivo activity against a wide spectrum of tumor cell lines [7,8,9]. They have been reported interesting antibiotic properties against a wide range of fungal and bacterial pathogens [5]. This review includes synthetic methodologies for the preparation of linear benzo[*g*]quinoxaline-5,10-diones derivatives and their structural analogs, as reported in the literature. Different strategies were found to give access to this class of compound, including:Condensations of 2,3-diamino-1,4-naphthoquinone with 1,2-dicarbonyl derivatives.Cycloaddition via generated in situ *N*-substituted-quinolinedione intermediates.Diels–Alder cyclocondensation reactions.

The second part of the review summarizes the common biological activities of benzo[*g*]quinoxaline-5,10-diones, with particular emphasis on their antitumor activity.

## 2. Discussion

### 2.1. Synthesis of Benzo[g]quinoxaline-5,10-diones and Their Uses in Heterocyclic Chemistry

#### 2.1.1. Synthesis from 2,3-Dichloro-1,4-naphthoquinone

Many series of substituted benzo[*g*]quinoxaline-5,10-diones at the C-2 and C-3 positions were reported. These methods involved several steps. The 2,3-dichloro-1,4-naphthoquinone **1** is the most used substrate for the preparation of 1,4-naphthoquinone heteroannulated to a quinoxaline (also named benzo[*g*]quinoxaline-5,10-diones) ring by multistep reactions synthesis (Scheme 1). The 2,3-dichloro-1,4-naphthoquinone **1** was used to obtain 2,3-diamino-1,4-naphthoquinone **2** (Scheme 2). This latter was condensed with a range of α-dicarbonyl compounds.

Reaction of 2,3-dichloro-1,4-naphthoquinone **1** with 2 equiv. of sodium azide produced a good yield of the diazido derivative, which was reduced with Na_2_S_2_O_4_ to give 2,3-diamino-1,4-naphtho-quinone **2** (**method a**—Scheme 2) [10,11,12,13,14]. 

As an alternative, a Gabriel reaction can also be used. Phthalimide potassium with alkyl halides allows the synthesis of primary amines. After alkylation, the product was cleaved by reaction with hydrazine (**method b**—Scheme 2). In this case, the reaction of compound **1** with 2 equiv. of phthalimide potassium afforded diphthalimido naphthoquinone, which was allowed to react with hydrazine hydrate to give compound **2** also named 2,2-diamino[1,4]-naphtoquinone (Scheme 2) [15,16].

Díaz et al. [17] synthesized compounds **3** as shown in Scheme 2 (**method c**) by melting **1** in ethanolic ammonia solution followed by acetylation. The acetylated compound **3** was suspended in an ammonia solution to obtain product **4**. Hydrochloric acid gas was then added in methanol to derivative **4** (Scheme 2).

Most of the examples herein involve cyclization reactions of 2,3-diaminonaphthoquinones **2** with symmetrical α-dicarbonyl compounds affording 2,3-disubstituted benzoquinoxaline- 5,10-diones as shown in Scheme 3 and Scheme 4 [18,19,20,21,22,23,24]. For example, compound **2** was allowed to react with bis(triisopropylsilyl)dialkynyl-1,2-dione in acetic acid to afford the corresponding compound **6a** [20]. It was reported that refluxing compound **2** with 1,2-di(1*H*-pyrrol-2-yl)ethane-1,2-dione in glacial acetic acid gave 2,3-di(1*H*-pyrrol-2-yl)benzo[*g*]quinoxaline-5,10-dione **6b** as illustrated in Scheme 3 [21].

Condensation of compound **2** with 1,4-dibromobutane-2,3-dione led to 2,3-bis(bromomethyl)benzo[*g*]quinoxaline-5,10-dione **6c** [18,22], which can be converted to its corresponding *bis*(chloromethyl) derivative by classical chlorination using lithium chloride [22]. Numerous reactions were also performed with unsymmetrical α-dicarbonyl compounds (Scheme 4, Table 1).

A non-exhaustive list of the substituents used in some of the most recent studies is presented in Table 1.

Reaction of **2** with oxalyl chloride afforded **7**, which directly reacted with thionyl chloride to give the corresponding dichloride, followed by a Gabriel reaction to obtain the corresponding diamino-compound **8**. Derivative **8** was then directly reacted with thionyl chloride or selenium oxychloride, to yield compound **9a** and **9b**, respectively (Scheme 5) [25].

Other polycyclic benzoquinoxalinediones were prepared from 2,3-diamino-1,4-naphthoquinone **2** by reaction with naphthalene-1,2-dione in 10% acetic acid and gave dibenzo[*a*,*i*]phenazine-8,13-dione **10** [26]. A similar reaction of **2** with phenanthrene-9,10-dione and 1,10-phenanthroline-5,6-dione produced tribenzo[*a*,*c*,*i*]phenazine-10,15-dione [16] **11a** and benzo[*i*]dipyrido[3,2-*a*:2′,3′-*c* ]phenazine-10,15-dione [26,27,28] **11b**, respectively (Scheme 6).

However, compound like **15b** (Scheme 7) could not be obtained by the reaction of the title compound 2 with the *o*-phenylene diamine. Another synthetic strategy to obtain diazaanthraquinones is the following. In a first step, the intermediate 2-aminoquinones are usually synthesized via a Michael-type reaction of amines with 1,4-naphthoquinone [29] itself or with 2,3-dichloro-1,4-naphthoquionone [30,31]. In a second synthesis step the *N*-arylaminoquinolinedione derivatives obtained are subjected to NaN_3_/moist DMF. Nakazumi et al. [32] obtained 7,10-dihydroxybenzo[*b*]phenazine-6,11-dione (X = OH and R = H) **15b** and 2-anilino-3-amino-5,8-dihydroxy-1,4-naphtoquinone **16b** by reacting 2-anilino-5,8-dihydroxy- 1,4-naphtoquinone **13b** with sodium azide, in 29% and 42% yield respectively (Scheme 7). The reaction of 2-benzylamino-3-chloro-1,4-naphthoquinone **13a** (X = H, R = H) [33] with sodium azide was also used to obtain benzo[*b*]phenazine-6,11-dione **15a** and **16a** as secondary products.

Kurban et al. [34] synthesized (2-chloro-3-tolylamino)naphthalene-1,4-dione (X = H, R = 2- or 3-CH_3_) from 2,3-dichloro-1,4-naphthoquinone **1**, which when reacted with sodium azide (NaN_3_) in moist DMF, afforded 1- or 2-methylbenzo[*b*]phenazine-6,11-dione **15c** or **15d** as the only isolated products (Scheme 7).

Most of the derivatives **13** can also be obtained through a nucleophilic substitution by reacting a substituted aniline with 2,3-dichloronaphthoquinone using CeCl_3_ as catalyst. Addition of this latter induces the formation of a complex between the carbonyl function and Ce^3+^. Thus, selective nucleophilic substitution takes place [35], with a chlorine atom replaced by a substituted aniline (68–90% yields with R = H; 4-OMe; 4-OEt; 2-F; 4-F; 3,4-diF; 2-OMe).

This approach is an easy way to synthesize tetracyclic linear diazaanthraquinone. The scope is limited by the variety of substituents when *N*-aryl is synthesized. Substituted anilines with electron-donating groups are quite reactive and afford high yields of the corresponding anilinoquinones. However, using either method with anilines substituted with electron-withdrawing groups gives quite low yields. As a consequence, these methods may not be used for the preparation of nitro substituted-2-(4-arylamino)-1,4-naphthoquinones. However, they can be obtained by an alternative two-step synthesis was developed. In the first step, 2-anilino-3-chloro-1,4-naphthoquinones were prepared by the classical Michael type addition-elimination reaction [33,36]. The phenyl group was then nitrated via direct electrophilic aromatic substitution in the second step.

The nucleophilic displacement reaction of compounds **17a**–**j** or 2-chloro-3-(*R*-anilino)- 1,4-naphthoquinones with sodium azide led to a vinylazide intermediates **18a**–**j** [35] (Scheme 8). Under heating, only few of these intermediates underwent thermal decomposition via intramolecular oxidative ring closure and yielded 6,11-dihydropyrido[2,3-*b*]phenazine-6,11-diones **19b**–**d** (Scheme 8).

Azides reduction (Staudinger reaction) is often limited by the formation of a secondary product (2-aryl- or 2-alkyl-amino-3-azido-1,4-naphthoquinone) from such reactions. Ring closure is promoted by azidonaphthoquinones bearing electron-rich substituents (R = OMe, OEt, OBut). The formation of arylaminonaphthoquinone derivatives is promoted by electron-withdrawing substituents. Based on Wasserman’s observations [37] regarding the mechanism of arylnitrene cyclisation could be explained in terms of singlet and/or triplet nitrene chemistry [35].

Azidonaphthoquinones having an electron-withdrawing substituent favor the generation of an open-shell nitrene with a strong biradical character, which undergoes hydrogen abstraction to give *N*-aminonaphthoquinone (Scheme 9). In contrast, azidonaphthoquinones having a strong electron-donor like O-R favor the generation of a highly reactive singlet nitrene which undergoes insertion to give a tetracyclic compound (Scheme 10).

When tricyclic dichloroquinone **20** reacts with arylamines, the intermediate mononitrogenated was obatained, from which further reaction with sodium azide afforded the expected linear pentacyclic naphtophenazinediones **21**–**23** in moderate to low yields (Scheme 11) with retaining the *p*-quinone system [38].

When the crude diazido compound, obtained from 6-nitro or 5-nitro-2,3-dichloro-1,4-naphthoquinones **24a** and **24b** was subjected to reduction with sodium dithionite followed by air oxidation, the triamino-1,4-naphthoquinone **25a** was obtained in 72% yield [32]. Compound **25a** was then condensed to glyoxal and yielded amino substituted benzo[*g*]quinoxaline-5,10-dione **26a** (92%) as shown in Scheme 12 [39].

The preparation of disubstituted derivatives of benzo[*b*]phenazine-6,11-quinones was reported [32]. Reduction of **15b** with sodium dithionite gave an intermediate compound which was reacted without purification with aqueous ammonia, methylamine or aniline to give, after subsequent oxidation, 7-10-diamino derivatives **27** (Scheme 13).

Interestingly, 2-arylamino-1,4-naphthoquinones **28a**–**d** treated with nitrosylsulfuric acid in acetic acid are converted to benzo[*b*]phenazine-6,11-dione oxides **29a**–**d**. Usually the 2-arylamino-3-chloro-1,4-naphthoquinones in reaction with HNO_3_/H_2_SO_4_ are nitrated on the arylamino group [36]. Thus, the reaction of aminoquinones **28a**–**d** with nitrosylsulfuric acid takes another pathway than that described in the literature. Compounds **29a**–**d** are obtained from aminoquinones **28a–d** in high yields and can be further modified. For example, phenazine *N*-oxide **29a** was reduced with hydrazine to benzo[*b*]phenazine-6,11dione **30a** in 89% yield (Scheme 14) [40].

It has been suggested, that the cyclization of 2-nitrosodiarylamines to the corresponding phenazine *N*-oxides [41] takes place upon treatment with nitrous acid. A mechanism involving radical cations is postulated for such cyclizations (Scheme 15), even though these fail to be detected. Notably, compounds **28** treated with nitrosylsulfuric acid in concentrated sulfuric acid failed to convert to products **29** even after prolonged stirring.

#### 2.1.2. Synthesis from 5,8-quinoxalinedione

The synthetic routes towards the construction of the benzo[*g*]quinoxaline-5,10-diones core from 5,8-quinoxalinedione were also studied [42,43].

4-Aminophenol **31** can be used as starting material. The diacetylation of **31** with acetic anhydride and triethylamine was achieved in 87% yield and was followed by nitration with fuming nitric acid to give mononitro compound **32** in 87% yield. The selective deacetylation of **32** was obtained using methanolic potassium carbonate at 0 °C (91%). Careful temperature control below 5 °C for nitration of **33** with 61% nitric acid gave single isomer 2,3-dinitro-compound **34** in 79% yield. Compound **34** was then hydrogenated in the presence of Raney Ni catalyst to afford 2,3-diamino-4-acetaminophenol, which was not isolated and reacted with glyoxal to form compound **35** in 79% yield. Compound **35** was readily converted to **36** with refluxing 2N sulfuric acid. Unisolated intermediate **36** reacted with sodium chlorate in hydrochloric acid solution at 0 °C to form 6,7-dichloro-5,8-quinoxalinedione **37** in 63% overall yield in eight steps, easily substituted with nucleophiles (Scheme 16). It is interesting to note that the title compound **37** was previously prepared in 3% yield in six steps [43].

Reaction of diaminoquinoxalinedione **38** with glyoxal in water [39] gave 5,10-pyrazino[2,3-*g*]quinoxalinedione **39a**. 2,3-Dialkyl-5,10-pyrazino[2,3-*g*]quinoxalinediones **39b**–**d** [44] were obtained by reacting diaminoquinoxalinedione with 1,2-diketones in 50% aqueous acetic acid. 2,3-Diphenyl-5,10-pyrazino[2,3-*g*]quinoxalinedione **39e** was obtained from the reaction of 6,7-diamino-5,8-quinoxalinedione **38** with benzil in ethanol using catalytic concentrated sulfuric acid (Scheme 17).

When a mixture of 6-bromo-7-methoxy-5,8-quinoxalinedione **40** and 1,1-diarylethylene derivatives was irradiated by Hg lamp at 300 W in presence of pyridine, naphtho[1,2-*g*]quinoxaline-7,12-dione derivatives **41** were isolated. Evidence was obtained that the photoreaction involved a photoinduced electron-transfer process (Scheme 18) [45].

The effect of electron-withdrawing and electron-donating groups on the reaction of 6,7-dichloro-5,8-quinoxalinedione **37** with dinucleophilic reagents was examined (Scheme 19). The reaction of **37** with 4-methoxy-1,2-phenylenediamine gave condensation compound corresponding to angulated annelated product **42**. The reaction with 4-nitro-1,2-phenylenediamine, because of the weak basicity of the amine, was not able to produce the condensation compound by ring closure; only **43** was isolated [46]. With aminopyridine, pyrido[1,2-*a*]imidazo[4,5-*g*]quinoxaline-6,11-dione **44** was obtained. Reaction mechanism can be either nucleophilic substitution or nucleophilic addition (Scheme 19) [46].

#### 2.1.3. Miscellaneous Reactions

The synthesis of 6,9-*bis*-[(aminoalkyl)amino]- substituted benzo[*g*]quinoxalines was described [47,48,49,50,51]. They were prepared by displacements (S_N_Ar) of the fluorides from 6,9-difluoro- substituted benzo[*g*]quinoxaline. The diacid was converted into the anhydride by refluxing in acetic anhydride or by treatment with DCC in THF. Friedel–Crafts acylation of 1,4-difluorobenzene with the anhydride in the presence of aluminium chloride led to keto acid **46**. Cyclodehydratation of **46** to **47** was obtained with fuming sulfuric acid at 140 °C. Addition of *N*,*N*-dimethylethylenediamine to a pyridine solution of **47** yielded **48a**. Fluoride displacements proceeded quite slowly and the mixture was stirred for several days to complete the *bis*-substitution. Shortening the reaction time would the mono-substituted analog to be isolated. **48b** and its mono-substituted analog were prepared by treatment of *N*-(*tert*-butoxycarbonyl)ethylenediamine in DMSO (Scheme 20).

The above synthetic pathway had the disadvantage to synthesize benzo[*g*]quinoxalinediones in which both distal side arms were the same. Krapcho et al. [48] accomplished subsequently the synthesis of regioisomeric dihalo-substituted heterocyclic quinone named 6-chloro-9- fluorobenzo[*g*]quinoxaline **56**. This molecule **56** can be used as starting material for the synthesis of dialkylamino-substituted heterocyclic quinones by fluoride and chloride on the C-6 and C-8 positions. The fluoride will undergo an S_N_Ar displacement at room temperature at a rate considerably more rapid than the chloride, on being treated with amine nucleophiles. The mono-chlorosubstituted compound being treated at a higher temperature with a different amine derivative (or any other nucleophilic species) will lead to the desired compounds. The synthesis of compound **56** is detailed on Scheme 21.

3-Aminopyrazine-2-carboxylic acid **49** was converted into 3-hydroxypyrazine-2-carboxylic acid **50** by diazotation and by heating the resulting diazonium salt in water. The corresponding acid was then converted into pyrazine ester **51** using dry hydrogen chloride in methanol. Chloroester **52** obtained by treatment with phosphorus oxychloride treated with the organozinc reagent derived from **53** in the presence of dichloro*bis*(triphenylphosphine)palladium and afforded **54**. This latter was hydrolyzed and led to the desired 6-chloro-9-fluorobenzo[*g*]quinoxaline **56**. This synthetic regioselective pathway avoids the Hayashi-type rearrangements [52] encountered during cyclizations of this family of keto acids (Scheme 21). A non-exhaustive list of diazabioisosteres of mitoxantrone **57a–l** already synthesized is summarized in Table 2.

Aryl- or alkylbenzo[*b*]phenazines **58a–c**, treated with a boiling solution of chromic acid/AcOH, resulted in the oxidation of benzophenazines to 2,3-aryl- or 2,3-alkyl-benzo[*b*]phenazine-6,11-diones **59a–c**, respectively (Scheme 22) [33,53].

2,3-Diaminonaphthalene was synthesized by successive chlorinations of 2,3-dihydroxynaphthalene **60** with POCl_3_ and amination using NH_3_/NH_4_Cl. The quinoxaline ring system was synthesized by reaction of *ortho*-phenylenediamine with α, β-dicarbonyl moieties. 2,3-Diphenylbenzo[*g*]quinoxaline **62** was obtained by oxidation of compound **61** with CrO_3_ in AcOH (Scheme 23) [54].

Microwave-assisted synthesis of fused pyrazolo[3,4-*b*]pyrazines **66** obtained by the reaction of orthoaminonitrosopyrazoles **65** and cyclic α-diketones was described (Scheme 24) [55]. Microwave irradiation is known to simplify and improve classic organic reactions, because it often leads to higher yields, cleaner and shorter reactions with precise control of its parameters [55].

A proposed mechanism [55] for the cyclocondensation reaction is outlined in Scheme 25. The reaction starts with a nucleophilic addition of the activated methylene to the nitroso group of pyrazole **65**, forming imino intermediate **67**. This addition is favored by the higher nucleophilicity of the activated methylene of the amino group of the pyrazole. Subsequently, intermediate **67** cyclizes via the remaining NH_2_ group with the terminal side chain carbonyl group (C=O) to form final pyrazolopyrazine **66** after dehydration (Scheme 25).

*p*-Dimethoxybenzene **69** can also be used as starting material. Nitration of **69** was performed in acetic acid with concentrated nitric acid followed by reduction with H_2_/Pd/C in ethyl acetate gave a mixture of compounds **70a**–**b** [56]. The regioisomers were separated in the next reaction step. Only compound **70a** gave quinoxaline **71** during the reaction with 2,3-butanedione, which could then be separated easily from the byproduct **70b**. Compound **71** was then transferred into compound **72** by reaction with AlCl_3_. Condensation of 2,3-dimethyl-5,8-dihydroxyquinoxaline **72** and *o*-phthalaldehyde gave 2,3-dimethylnaphtho[2,3-*g*]quinoxaline-5,12-dione **73** in 70% yield [57] as shown in Scheme 26.

#### 2.1.4. Diels–Alder Reactions

Diels–Alder reactions of quinones with a variety of polarized dienes are efficient methods to synthesize highly substituted diazaanthraquinones. Quinoxaline quinones can serve as useful dienophiles in [4+2]-cycloaddition reactions. They were obtained via a Hinsberg reaction from the condensation of *o*-diamine **75** with glyoxal [58] or 2,3-butanedione [58,59] followed by ceric ammonium nitrate (CAN) oxidation to afford **76a** and **76b** (68% and 85% overall yield, respectively) as shown in Scheme 27.

The synthetic pathway starts with the nitration of 1,4-dimethoxybenzene **74**. The obtained mixture of dinitro compounds isomers can be reduced by different methods [43,60] or separated before the reduction step [61,62]. With NO_3_/O_3_ in dichlomethane at 0 °C [61], a good yield (81%) was obtained in the mixture of compounds **75a** (1,4-dimethoxy-2,3-dinitrobenzene) and **75b** (1,4-dimethoxy-2,5-dinitrobenzene), with no selectivity (54/46 respectively). The NO_2_/BF_3_ in dimethoxyethane at −50 °C [63] led to regioisomers **74a**/**74b** in 76% yield with an excellent regioselectivity (100/0). A step-by-step reduction was found to be advantageous, allowing *o*-amino **75** to be easily isolated in good yield. Such a stepwise procedure avoids the separation of **75a**/**75b** or the corresponding diamines (Scheme 27) [60].

Preparation of quinoxaline-5,8-dione **76a** is reported in the literature [64]. Benzo[*g*]quinoxaline-5,10-dione derivatives bearing 7-dialkylaminomethyl were synthesized based on the Diels–Alder reaction of quinoxaline-5,8-dione **76a** with isoprene [65], as outlined in Scheme 28. Starting material **76a** was treated with isoprene to give the cycloaddition adduct, which was directly aromatized by aerial oxidation in 5N ethanolic KOH under reflux to give 7-methylbenzo[*g*]quinoxaline-5,10-dione **78**. Intermediate **78** was then treated by *N*-bromosuccinimide (NBS) and a catalytic amount of benzoylperoxide in anhydrous 1,2-dichloroethane under reflux for 48 h with irradiation by tungsten lamp to give bromomethyl product **79** in 30% yield (Scheme 28).

Starting material **76a** was treated with 2,3-dimethylbutadiene to give the cycloaddition adduct, which was directly aromatized by aerial oxidation in ethanolic KOH under reflux to give 6,7-dimethylbenzo[*g*]quinoxaline-5,10-dione **82** [66,67]. The required intermediate **82** was treated with *N*-bromosuccinimide (NBS) and a catalytic amount of benzoylperoxide under irradiation to give the corresponding bisbromomethyl product in 42% yield. Target compounds **83** containing alkyl or cycloalkyl substituents were synthesized by direct substitution reaction of the *bis*bromomethyl compound with the corresponding alkylamine, and afforded alkyl- substituted triazacyclopenta[*b*]anthracene-5,10-dione derivatives in 30% to 81% yield (Scheme 29).

Further reactions of compounds **82** with 18% HNO_3_ in a Parr-type A-30397 titanium pressure reactor afforded the corresponding dicarboxylic acid **84**. The crude product heated to reflux in SOCl_2_, and subsequent treatment of this crude with a number of alkyl- or arylamines, led to compounds **85a**–**j** in 16% to 31% global yields (Scheme 30) [67].

7,8-Dimethylbenzoquinoxalinediones are useful as starting materials for the synthesis of tricyclic quinones through the Diels–Alder reaction.

Cycloadditions with cyclic dienes are expected to occur with formation of a 1:1-cycloadduct, followed by tautomerization and ready oxidation. This yields a bridged ring system which can undergo thermal elimination of ethylene to afford diazaanthraquinone [51]. Cycloadditions of 1,3-cyclohexadiene with the other heterocyclic quinones also yielded initial 1:1-cycloadducts **86**, which were isolated directly from the reaction mixture. Conversion into the oxidized quinone with silver oxide and thermal elimination of ethylene afforded diazaanthraquinones in high yields (Scheme 31).

2,3-Dimethylquinoxaline-5,8-dione **76b** was reacted with silyloxyl dienes (Scheme 32) in refluxing benzene, giving trihydroxy compound **89** (90%). Acetylation of trihydroxy compound **89** afforded the corresponding acetate which, after oxidation, was converted into quinone **91**. As shown in Scheme 32, Danishefsky’s diene (*trans*-1-methoxy-3-[(trimethylsilyl)oxy]-l,3-butadiene) easily underwent cycloaddition with the quinone **76b** to directly afford oxidized cycloadduct **91** (71%).

The total synthesis of pyrazine analogs of 1,1-deoxydaunomycin was reported. Treatment of sodium salts generated from tetrahydrohomophthalic anhydride (Scheme 33) with derivative **76b** gave cycloadducts **92** and **93**, regioselectively [68]. These adducts were converted to pyrazine analogs of 1,1-deoxydaunomycin **94** and **95**.

Treatment of the sodium salts cited above with mercury(II) oxide and diluted sulfuric acid, followed by bromination with bromine and AIBN, gave *cis*-diol **92** in 27% yield. The respective configurations of **92** and **93** were determined from proton nuclear magnetic resonance (1*H*-NMR) spectral data [69]. Condensation of racemic **92** with tetrahydropyrane derivative followed by base hydrolysis gave α-glycosides **94** and **95** as an inseparable 1:1 mixture of two diastereomers in 48% yield (Scheme 33).

A useful approach to the synthesis of anthracyclinone derivatives (Scheme 34) is the Diels–Alder using quinone **98** [70]. The latter was obtained (88% yield) by oxidation of 1,4-diol **97a** with lead tetraacetate, or directly from 1,4-dimethoxy precursor **97b** [71] with silver oxide in nitric acid [72] (40% yields). Successful oxidation of **96** (*via* the di-*O*-acetate) to di-*N*-oxide was reported [71].

Quinone **98**, unoxidized at the nitrogen atom, proved to be a good dienophile with reactive 1-methoxy-3-(trimethylsiIyloxy)-l,3-butadiene [73]. The reaction occurred at room temperature to give the normal adduct **99** in 96% yield. Adduct **99**, prone to aromatization, gave elimination product **100** [70].

### 2.2. Antitumor Activities

Tricyclic diazaquinones exhibit strong anticancer activity [48]. One of the cytostatic action mechanisms of coplanar polycyclic compounds is their intercalation with human DNA. This caused enzymatic blockade and reading errors during the replication process [74]. Compounds having three to four coplanar rings, appear to give the optimal intercalation. More annelated heterocyclic quinones were reported to increase antitumor activity [75]. The electrochemical properties of quinone compounds are obviously very important for their bioreduction to semiquinone and/or hydroquinone. The replacement of two carbons at the phenyl ring of the 1,4-naphthoquinone core by two nitrogen atoms increased the oxidant nature of the molecules in accordance with both redox potential and substrate efficiencies [76].

2,3-Diethyl-5,10-pyrazino[2,3-*g*]quinoxalinedione **39d**, prepared from 6,7-dichloroqui- noxaline-5,8-dione **37** in 59% overall yield in three steps [39] exhibited potent cytotoxic activity against human gastric adenocarcinoma cells (IC_50_ = 1.30 and 7.61 µM, respectively). Both compounds bearing bulky side chains are supposed to interact with DNA and form a stable complex (Figure 2).

Many heterocyclic quinones act as topoisomerase inhibitors via DNA-intercalation. Topoisomerases are DNA-modifying enzymes essential to the control of DNA topology. They are involved in all cellular processes (replication, transcription, chromatin condensation and recombination) in which the topology of the DNA molecule must be changed without changing its chemical structure. Some pyridophenazinediones [77] showed potent activity against human stomach cancer cells (Figure 3). The possible mechanism of this action is suggested to be a DNA topo I and topo II inhibition. The best representatives are compounds **101** (IC_50_ = 0.06 µM on SK-OV-3), prepared from 6,7-dichloroquinoline-5,8-dione in 38% overall yield in two steps, and **102** (IC_50_ = 0.06 µM on XF-498), prepared from 6,7-dichloroquinoline-5,8-dione in 23% overall yield in two steps.

Giorgi-Renault prepared benzoquinoxalinediones and examined their antitumor activities [18]. 2,3-*Bis*(bromomethyl)-5,10-benzo[*g*]quinoxalinedione **6c**, prepared from 2,2-diamino[1,4]-naphto- quinone **2** in 75% yield in one step, exhibited cytotoxicity and was highly active, especially against sarcoma (Figure 4).

The tumor suppressor p53 is a central mediator of apoptosis from chemically induced stress. Doxorubicin causes activation of p53 in both diploid and tetraploid cells due to a lack of polyploid cell line-specific selectivity. Recently, 2,3-diphenyl-1,4-diazaanthraquinone (DPBQ) **62** was proven to be a selective lead compound for the treatment of high-ploidy breast cancer, which activates p53 and triggers apoptosis of tumor cells (Figure 5) [78]. This latter compound appears to be limited to high-ploidy cell types with intact p53. It does not inhibit topoisomerase or bind DNA. Mechanistic analysis demonstrates that DPBQ **54**, obtained from the National Cancer Institute (NCI) elicits a hypoxia gene signature and its effect is replicated, in part, by enhancing oxidative stress.

A number of 6,7-modified 5,8-quinoxalinedione derivatives containing nitrogen, sulfur and oxygen exhibited cytotoxic effects on human lung, gastric and colon adenocarcinoma cells when compared with *cis*-platin and adriamycin, commonly used anticancer drugs [79].

The cytotoxicity of 6,7-modified-5,8-quinoxalinedione derivatives and heterocyclic quinoxaline derivatives containing nitrogen (compounds were obtained from the NCI Open Compound Repository, Drug Synthesis and Chemistry Branch, NCI) was evaluated in vitro using an MTT assay on human lung adenocarcinoma cells (PC 14), human gastric adenocarcinoma cells (MKN 45), and human colon adenocarcinoma cells (colon 205). Pyrido[1,2-*a*]imidazo[4,5-*g*]quinoxaline-6,11-dione **44** was markedly cytotoxic against MKN **45** compared with adriamycin and cis-platin used as reference drugs. The IC_50_ value of compound **44** was 0.073 µM while those of adriamycin and *cis*-platin were 0.12 µM and 2.67 µM, respectively. In this study, the relationship between structure, redox cycling, and cytotoxicity in the MCF-7 and HL-60 cell lines was investigated (Figure 6) [80].

To evaluate their reactivity for bioreductive activation, the levels of free radicals under aerobic conditions were quantified. Direct ESR evidence of formation was provided. The data suggest that good levels of free radicals are generated in the HL-60 (e.g., the HL-60 myeloid leukemia) and MCF-7 cells (e.g., the MCF-7 breast carcinoma) by compound B. Group B compounds showed good redox cycling in both cell lines (HL-60 and MCF-7 cells). They were cytotoxic in the MCF-7 cell at concentrations down to 10 μM (Figure 7).

## 3. Conclusions

Despite the effort made to design benzo[*g*]quinoxalinedione compounds, there are few efficient synthetic approaches, especially green chemistry approaches. 1,4-diazaanthraquinones stand out because of their wide range of biological activities, including anti-cancer activity. However, exploration of these activities has been limited. These drugs offer a larger repertoire of activities in cancer cells than is currently exploited. The heterocyclic quinones that appeared to be very promising in vitro proved ineffective in studies. One reason for this failure is insufficient understanding of the mechanisms of action of these compounds, in part due to the numerous modes of action of these families of derivatives. The versatility of heterocyclic quinones containing a quinoxaline core and their potential to be selectively toxic to tumor cells hold great promise for anticancer therapy.

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
