# Peer review of "A Survey of Synthetic Routes and Antitumor Activities for Benzo[g]quinoxaline-5,10-diones"

_molecules, 2020, doi:10.3390/molecules25245922_

Round 1

Reviewer 1 Report

The subject is quite interesting, the cited references are accurate, but the material is presented in such a way that one does not benefit at all from this review: 1. the text is in some parts difficult to follow (see, for example, the discussion about compounds of type 15 and 16 on pages 5 and 6 is intercalated by compounds of type 17... - this part should be reorganized to bring clarity and fluidity); 2. there are puzzling schemes (scheme 1 is just mentioned in the text, nothing is actually said about it; scheme 4 is highly confusing - some compounds are numbered and some are not; and even more confusing is scheme 7 with the same style, except that this time compounds 15 a-e, for example, are named and described, only to find ourselves in the next page with a discussion on compounds 15 e-i, not named in the scheme; why have 17a if it is only one compound of this kind and which is compound 17'; and what about compound 49 between compounds 28 and 29? etc.); pay attention also to keeping the same settings for drawing the structures throughout the paper! 3. missing references (fragment at page 7, lines 153-156 has no literature reference and in fact is taken word by word from reference 77) or doubled (Synthesis 1983); 4. errors (discussion at page 8, lines 183-189 misinterpreted the data in literature reference; table 1 on compounds 57 (though the structure of compounds 57 is not presented anywhere, we can only presume it is that in the header of table 1) has no literature reference in the text, but instead, in the body of the table there are some odd references, most probably kept as such from the original research paper); 5. wrong names for some compounds and so on. Moreover, a more critical approach in some parts describing the syntheses (e.g. more comparative yield references) would be welcome.   The paper has potential, but needs to be revised thoroughly and rewritten in some parts, as it cannot be considered for publication as it is now.

Author Response

We are grateful to the reviewers for the positive and constructive comments, which helped to improve the quality of the manuscript. Herewith, we are submitting point to point response to comments/queries posed by the reviewer. We have incorporated all the suggestions made by the reviewer in the revised manuscript.

  1. the text is in some parts difficult to follow (see, for example, the discussion about compounds of type 15 and 16 on pages 5 and 6 is intercalated by compounds of type 17... - this part should be reorganized to bring clarity and fluidity);

We have made necessary corrections as per the reviewer’s comment. The schemes and the text have been modified.

  1. there are puzzling schemes (scheme 1 is just mentioned in the text, nothing is actually said about it; scheme 4 is highly confusing - some compounds are numbered and some are not; and even more confusing is scheme 7 with the same style, except that this time compounds 15 a-e, for example, are named and described, only to find ourselves in the next page with a discussion on compounds 15 e-i, not named in the scheme; why have 17a if it is only one compound of this kind and which is compound 17'; and what about compound 49 between compounds 28 and 29? etc.); pay attention also to keeping the same settings for drawing the structures throughout the paper! 

We have made necessary corrections. The numbering of the compounds has been changed as the scheme 8.

The text corresponding to scheme 1 has been modified.

The scheme 4 has been divided in two schemes and a table has been added.

  1. missing references (fragment at page 7, lines 153-156 has no literature reference and in fact is taken word by word from reference 77) or doubled (Synthesis 1983); 

The Scheme 11 has been replaced and its corresponding text also as per the other reviewer’s comment (It was not corresponding to the title).

Reference has been added and the bibliographic part has been modified.

The reference (synthesis 83) was not doubled.  The same authors published in the same journal the same year.

  1. errors (discussion at page 8, lines 183-189 misinterpreted the data in literature reference; table 1 on compounds 57 (though the structure of compounds 57 is not presented anywhere, we can only presume it is that in the header of table 1) has no literature reference in the text, but instead, in the body of the table there are some odd references, most probably kept as such from the original research paper); 

We have made necessary corrections. The schemes and the text have been modified and the odd references corrected.

  1. wrong names for some compounds and so on. Moreover, a more critical approach in some parts describing the syntheses (e.g. more comparative yield references) would be welcome. 

We have made necessary corrections (lines 104-105, 205-206, 244-245, 318-318)

Reviewer 2 Report

The manuscript by Giuglio-Tonolo et al. reports the synthetic approaches to benzo[g]quinoxaline-5,10-diones and the antitumor activities of the most promising compounds.

The pharmacological activities are briefly named in the first part of the introduction and cannot be considered exhaustive, while in the second part of the discussion (paragraph 2.2) only the anticancer activity is reported. For this reason, “antitumor activity” and not “biological activity” should be reported in the title.

In all the schemes, the molecule number should be placed under the molecule for easier understanding of, as well as the names of the molecules in Figure 1.

Scheme 3 which corresponds to method c of scheme 2 could be reported in the same scheme 2.

In scheme 4 substituents R1 and R2 are only named, as for example in scheme 30. It would be useful to insert a table with substituents and bibliographic references.

The paragraph from line 120 to 143 is very confused and difficult to read even with reference to diagrams 8 and 9. The text could be reorganized in a more linear way.

Compounds 23 a -d are pyrido [2,3-g] quinoxaline-5,10 (4aH, 10aH)-dione and not [g]quinoxaline-5,10-diones.

It would be appropriate to report the percentage yields of the final products and indicate which compounds were the most promising and submitted to biological evaluation.

In paragraph 2.2 only the antitumor activity of some compounds is reported.

In the conclusion, green chemistry is mentioned without having highlighted this aspect in the discussion. It would be appropriate to discuss this in paragraph 2.1 and then insert it in the conclusion. The same is for the limitation of these compounds in biological studies due to lack of understanding of the biological mechanism. It should be prior discussed in paragraph 2.2.

According to my considerations, this paper needs major revision.

These items should be addressed prior to publication.

Author Response

We are grateful to the reviewers for the positive and constructive comments, which helped to improve the quality of the manuscript. Herewith, we are submitting point to point response to comments/queries posed by the reviewer. We have incorporated all the suggestions made by the reviewer in the revised manuscript.

The pharmacological activities are briefly named in the first part of the introduction and cannot be considered exhaustive, while in the second part of the discussion (paragraph 2.2) only the anticancer activity is reported. For this reason, “antitumor activity” and not “biological activity” should be reported in the title.

We have made necessary corrections on the title as per the reviewer’s comment.

  • In all the schemes, the molecule number should be placed under the molecule for easier understanding of, as well as the names of the molecules in Figure 1.

We have made necessary corrections in all the schemes as per the reviewer’s comment.

  • Scheme 3 which corresponds to method c of scheme 2 could be reported in the same scheme 2.

We have made necessary corrections.

  • In scheme 4 substituents R1 and R2 are only named, as for example in scheme 30. It would be useful to insert a table with substituents and bibliographic references.

A table has been added for scheme 4 and 5 which renamed scheme 3 and 4.

For scheme 30, as there is only one substituent R on depicted molecules, a table was not added, but the scheme was modified, and yields added for better clarity.

  • The paragraph from line 120 to 143 is very confused and difficult to read even with reference to diagrams 8 and 9. The text could be reorganized in a more linear way.

We have made necessary corrections as per the reviewer’s comment. The diagrams and the text have been modified.

  • Compounds 23 a -d are pyrido [2,3-g] quinoxaline-5,10 (4aH, 10aH)-dione and not [g]quinoxaline-5,10-diones.

We have made necessary corrections and scheme 11 has been changed.

  • It would be appropriate to report the percentage yields of the final products and indicate which compounds were the most promising and submitted to biological evaluation.

The percentage yields of the final products have been added when it was possible.

The core benzo[g]quinoxaline-5,10-dione is necessary for the antitumor effects. It is complex to indicate which compound is the most promising since their mode of actions are very different and they have not the same targets. The antitumor activities are cell line-specific.

  • In paragraph 2.2 only the antitumor activity of some compounds is reported.

The title 2.2 has been modified.

  • In the conclusion, green chemistry is mentioned without having highlighted this aspect in the discussion. It would be appropriate to discuss this in paragraph 2.1 and then insert it in the conclusion. The same is for the limitation of these compounds in biological studies due to lack of understanding of the biological mechanism. It should be prior discussed in paragraph 2.2.

The green chemistry approach has been detailed in paragraph 2.1.

In our opinion the lack of understanding of the biological mechanism resulted to the numerous modes of action of these family of derivatives which was already discussed in paragraph 2.2. It has been highlighted in the conclusion.

Round 2

Reviewer 1 Report

The paper has gained a lot in readability. There are still some errors in reagents, compound names, references etc., I have attached direct comments or blue highlights on the revised .pdf file. Again, please verify that all schemes have the same ChemDraw document settings!

Author Response

We are grateful to you for your help to improve the quality of the manuscript.

We have incorporated all the suggestions in the revised manuscript. I have yellow highlights the corrections on the revised word document named "molecules CM4".

Best regards

Reviewer 2 Report

I appreciated the revision of the paper and the authors' response.

The revision meets all my requests and should be published in the present form.

Author Response

(The authors gave the same response as above.)
